# Comparison of the Effects of Botulinum Toxin Doses on Nerve Regeneration in Rats with Experimentally Induced Sciatic Nerve Injury

**DOI:** 10.3390/toxins15120691

**Published:** 2023-12-08

**Authors:** Seokjoon Hwang, Minsu Seo, Tae Heon Lee, Ho Jun Lee, Jin-woo Park, Bum Sun Kwon, Kiyeun Nam

**Affiliations:** Department of Physical Medicine & Rehabilitation, Dongguk University College of Medicine, Goyang 10326, Republic of Korea; seokjoon.h@dumc.or.kr (S.H.); seoms87@dumc.or.kr (M.S.); taeheon320@dumc.or.kr (T.H.L.); hjrhee1@dumc.or.kr (H.J.L.); jinwoo_park@dumc.or.kr (J.-w.P.); bskwon@dumc.or.kr (B.S.K.)

**Keywords:** botulinum toxin, peripheral nerve injury, neural regeneration, functional recovery

## Abstract

This study was designed to compare the effects of various doses of botulinum neurotoxin A (BoNT/A) on nerve regeneration. Sixty-five six-week-old rats with sciatic nerve injury were randomly allocated to three experimental groups, a control group, and a sham group. The experimental groups received a single session of intraneural BoNT/A (3.5, 7.0, or 14 U/kg) injection immediately after nerve-crushing injury. The control group received normal intraneural saline injections after sciatic nerve injury. At three, six, and nine weeks after nerve damage, immunofluorescence staining, an ELISA, and toluidine blue staining was used to evaluate the regenerated nerves. Serial sciatic functional index analyses and electrophysiological tests were performed every week for nine weeks. A higher expression of GFAP, S100β, GAP43, NF200, BDNF, and NGF was seen in the 3.5 U/kg and 7.0 U/kg BoNT/A groups. The average area and myelin thickness were significantly greater in the 3.5 U/kg and 7.0 U/kg BoNT/A groups. The sciatic functional index and compound muscle action potential amplitudes exhibited similar trends. These findings indicate that the 3.5 U/kg and 7.0 U/kg BoNT/A groups exhibited better nerve regeneration than the 14 U/kg BoNT/A and control group. As the 3.5 U/kg and the 7.0 U/kg BoNT/A groups exhibited no statistical difference, we recommend using 3.5 U/kg BoNT/A for its cost-effectiveness.

## 1. Introduction

Peripheral nerve injury is an essential medical condition because it can impair patient function by increasing the risk of secondary disabilities [1]. These injuries are frequently caused by trauma, lacerations, or fractures. Moreover, ischemia, traction, electric shock, and vibration can play a role [2]. An estimated 2.8–5.0% of trauma admissions in North America and Europe involve these injuries [3]. As neural tissue injuries have a significant impact on patients’ quality of life, their regeneration has long been of great interest in medicine [4]. Numerous attempts have been made to improve the healing of injured peripheral nerves [5]. Peripheral nerve injuries are typically treated surgically. Primary end-to-end repair and nerve grafts are two surgical treatment options [6]. Nonsurgical therapeutic methods, such as physical therapy, hydrotherapy, electrical stimulation, and medicine, have received the most attention. Recently, several studies have reported that botulinum neurotoxin A (BoNT/A) is effective in nerve regeneration [7,8,9,10].

BoNT/A is used for several indications including cosmetics, muscle spasticity, migraines, dystonia, and axillary hyperhidrosis. It blocks acetylcholine from being released by cholinergic nerve terminals through the cleavage and degradation of a synaptosomal-associated protein of 25 kDa, which reduces muscle contractility [3]. BoNT/A has well-established therapeutic applications that are constantly growing, particularly in the management of peripheral nerve damage. It is assumed that BoNT/A directly stimulates axonal regeneration in injured peripheral nerves, leading to neurological recovery [7]. Following complete compression injury to the sciatic nerve, Cobianchi showed that a single injection of botulinum toxin into the nerve accelerated axonal growth and increased the number of regenerated myelinated fibers [7]. Previous studies have shown that BoNT/A affects the improvement of electrophysiological recovery and promotes neural sprouting [3,11]. A recent study by Luvisetto et al. reported that BoNT/A can modulate astrocyte activity, which plays a crucial role in regulating neuronal function. BoNT/A can inhibit the release of proinflammatory molecules from astrocytes, which may have implications for the treatment of neuroinflammatory disorders [8].

However, there has been no research on the most appropriate dose of BoNT/A that produces optimal neurological recovery. This study aimed to investigate and compare neural recovery after treatment with various doses of onabotulinum neurotoxin type A in rats with sciatic nerve injury produced experimentally. An enzyme-linked immunosorbent assay (ELISA), immunofluorescence staining, and toluidine blue staining were used in the experiments. The sciatic functional index (SFI) and electrophysiological tests were used to assess functional improvement.

## 2. Results

### 2.1. Effect of BoNT/A on Injured Sciatic Nerves’ Schwann Cell Activity (GFAP, S100β)

Immunofluorescence staining revealed increased glial fibrillary acid protein (GFAP) and astroglial calcium-binding protein S100β (S100β) expression in all BoNT/A (3.5, 7.0, and 14 U/kg) groups. Among the BoNT/A-injected groups, the expression in the 14 U/kg BoNT/A group appeared to be the least enhanced (Figure 1A and Figure 2A).

The ELISA showed significantly higher GFAP expression levels (Figure 1B) in the 3.5 and 7.0 U/kg BoNT/A groups compared to the 14 U/kg BoNT/A and control groups at week 3. By the ninth week, the 7.0 U/kg, 3.5 U/kg, control, and 14 U/kg BoNT/A groups had sequentially significantly higher expression rates. The ELISA expression levels of S100β (Figure 2B) were significantly higher in the 3.5 U/kg and 7.0 U/kg BoNT/A groups than in the 14 U/kg BoNT/A and control groups at weeks 6 and 9. No difference was seen within the 3.5 U/kg and 7.0 U/kg BoNT/A groups at weeks 3, 6, and 9. The 14 U/kg BoNT/A and control groups did not show any significant differences at weeks 6 and 9.

### 2.2. Effect of BoNT/A on Injured Sciatic Nerves’ Axon Regeneration (GAP43, NF200)

When compared to the control group, the expression levels of growth-associated protein 43 (GAP43) and neurofilament 200 (NF200) in all BoNT/A (3.5 U/kg, 7.0 U/kg, and 14 U/kg) groups were markedly higher (Figure 3A and Figure 4A).

The ELISA showed greater expression levels of GAP43 and NF200 in the 3.5 and 7.0 U/kg BoNT/A groups than in the 14 U/kg BoNT/A and control groups at weeks 6 and 9 (
Figure 3B and Figure 4B). Comparing GAP43 within the BoNT/A groups, the 14 U/kg BoNT/A group did not significantly differ from the 3.5 and 7.0 U/kg BoNT/A groups at week 3. However, at weeks 6 and 9, the 14 U/kg BoNT/A group showed a significant decrease in expression levels. The GAP43 expression levels in the 14 U/kg BoNT/A group were considerably lower than in the control group at week 9 (*p* < 0.05). For the NF200 expression levels, the 14 U/kg BoNT/A and control groups showed a significant difference only at week 9. Throughout the whole experiment, no statistically significant difference was seen among the 3.5 and 7.0 U/kg BoNT/A groups.

### 2.3. Effect of BoNT/A on the Nerve Growth Factors of Injured Sciatic Nerves (BDNF, NGF)

Higher brain-derived neurotrophic factor (BDNF) and nerve growth factor (NGF) expression in the BoNT/A group was observed with immunofluorescence staining (Figure 5A and Figure 6A).

BDNF expression, analyzed in the ELISA, was significantly upregulated in the BoNT/A groups compared to the control group at weeks 6 and 9 (*p* < 0.05) (Figure 5B). At week 3, the expression levels in the 14 U/kg BoNT/A group were higher than in the control group, but the difference was not statistically significant. However, the NGF expression in the BoNT/A group was significantly higher than in the control group at all experimental weeks (Figure 6B). The increment in the 14 U/kg BoNT/A group was less than that of the 3.5 and 7.0 U/kg BoNT/A groups. The 3.5 and 7.0 U/kg BoNT/A groups showed no statistically significant difference in the expression levels of either factor.

### 2.4. Toluidine Blue Staining

Images of the embedded sciatic nerve sections stained with toluidine blue were analyzed in three categories (Figure 7A).

The average area of regenerated axons was significantly larger in the 3.5 and 7.0 U/kg BoNT/A groups than in the 14 U/kg BoNT/A and control groups (Figure 7B). At weeks 6 and 9, the 14 U/kg BoNT/A group had a larger area than the control group. The 3.5 U/kg BoNT/A group had a significantly larger area than the 14 U/kg BoNT/A group at week 3, but the 7.0 U/kg BoNT/A group did not.

When comparing the mean diameter of the myelin sheath, the trend was nearly identical to that of the average area. The only difference was that the 7.0 U/kg BoNT/A group showed a significant difference compared to the 14 U/kg BoNT/A group at week 3 (Figure 7C).

The remyelination degree of the regenerated axons (G-ratio) was significantly lower in the 3.5 and 7.0 U/kg BoNT/A groups than in the 14 U/kg BoNT/A and control groups. The 3.5 and 7.0 U/kg BoNT/A groups showed close convergence to the ideal value of the G-ratio by week 9 [12]. Within the groups, the 3.5 and 7.0 U/kg BoNT/A groups did not show significant differences (Figure 7D).

### 2.5. Effects of BoNT/A on Functional Recovery after Sciatic Nerve Injury 

#### 2.5.1. Compound Muscle Action Potential Amplitude 

Both the experimental and control groups’ compound muscle action potential (CMAP) amplitudes gradually increased, according to the electrophysiological results, suggesting that their motor function had recovered (Figure 8A). Compared to the control group, all BoNT/A (3.5, 7.0, and 14 U/kg) groups showed a statistically significant increase in amplitude after two, three, and four weeks (*p* < 0.05). Comparing the efficiency between the BoNT/A groups, a larger CMAP was observed in the 3.5 U/kg BoNT/A group than in the 14 U/kg BoNT/A group from the eighth week. The 7.0 U/kg BoNT/A group had a greater CMAP than the 14 U/kg BoNT/A group only in the ninth week. Although all BoNT/A (3.5, 7.0, and 14 U/kg) groups showed significant improvement, the CMAP was still significantly lower than that in the sham group by week 9. This finding suggests that normal neurophysiology was not achieved.

#### 2.5.2. Sciatic Functional Index

In the 3.5 and 7.0 U/kg BoNT/A groups, the SFI results were similar to those of the electrophysiological study (Figure 8B). The results were significantly superior to those in the control group after the second and third weeks (*p* < 0.05). However, the 14 U/kg BoNT/A group was found to be significantly different from the control group only in the ninth week. While comparing the effectiveness of the BoNT/A groups, it was found that, starting in the eighth week, both the 3.5 and 7.0 U/kg BoNT/A groups had greater SFI results than the 14 U/kg BoNT/A group. The SFI was not significantly different from the sham group after week 8 for the 3.5 U/kg BoNT/A group and week 9 for the 7.0 U/kg BoNT/A group. This indicates the recovery of normal gait patterns. In all results throughout the experimental period, the 3.5 and 7.0 U/kg BoNT/A groups showed no significant difference in CMAP and SFI.

## 3. Discussion

Our study is the first to investigate the neuroregenerative effects of varying doses of BoNT/A. The overall purpose of this investigation was to determine the optimal doses of BoNT/A for neural regeneration. Our previous study showed a superior effect of BoNT/A than the normal saline control group in nerve regeneration and functional recovery after peripheral nerve injury in rats [10]. The stimulation of Schwann cells and axonal regrowth after BoNT/A appeared to improve functional recovery, according to the electrophysiological, clinical behavioral data, and neural tissue analysis results presented in this study. Higher expression levels of proteins for Schwann cell activities, axonal regeneration, and nerve growth were more pronounced in the 3.5 and 7.0 U/kg BoNT/A groups compared to the other groups. Furthermore, we found that the 3.5 and 7.0 U/kg BoNT/A groups showed a larger area and thicker myelin sheath through toluidine blue staining. Moreover, by week 9, the G-ratio was nearly 0.6, which is the optimal value [12]. In addition to morphology, the electrophysiological findings confirmed that the 3.5 and 7.0 U/kg BoNT/A groups showed a larger CMAP than the control group. In the functional evaluation, the SFI showed trends identical to those of the other results. Although the 3.5 and 7.0 U/kg BoNT/A groups did not reach the CMAP amplitude of the sham group on the NCS at week 9, the SFI fully recovered and was similar to the normal gait pattern of the sham group. These results indicate that the 3.5 and 7.0 U/kg BoNT/A groups have a greater neuroregenerative effect morphologically, electrophysiologically, and functionally than the 14 U/kg BoNT/A and control groups.

Botulinum toxins are produced by anaerobic bacteria of the genus Clostridium under seven different serotypes, referred to as BoNT/A to BoNT/G, and include several variants [13,14]. Among those serotypes, the clinically used botulinum toxin is type A and type B. In a 2018 study by Finocchiaro et al., botulinum toxin type B did not elicit functional recovery after peripheral nerve injury. Therefore, botulinum toxin type A was used in this study [15].

Recent research has elucidated the mechanisms underlying the neuroregenerative effects of BoNT/A after peripheral nerve injury. Some of BoNT/A’s less well-known functions have been shown in earlier research, such as its ability to speed up nerve regeneration after axotomy, nerve crush, and chronic constriction injury. Franz et al. reported the effects of BoNT/A preconditioning on reinnervation [16]. Adlet et al. suggested that BoNT/A may accelerate nerve regeneration by improving blood flow and upregulating angiogenesis [17]. A study by Finocchiaro et al. describes that mast cells and macrophages are further activated to speed up the removal of myelin debris and promotes nerve regeneration after BoNT/A injection [15]. We focused on the role of Schwann cells in peripheral nerve regeneration. Nerve regeneration is predominantly linked to enhanced Schwann cell proliferation, transformation to the repair phenotype, and mast cell and macrophage activation [17]. Spinal glial cell activation and the consequent release of pro-inflammatory factors can be inhibited by BoNT/A [8]. BoNT/A can interact with glial cells such as Schwann cells and oligodendrocytes, which are responsible for rebuilding the myelin sheath of neural axons, possibly resulting in a gradual recovery of myelin function following injury [8]. Schwann cells produce neurotrophic factors to provide a favorable environment for axon regeneration and create guidance tracks (bands of Büngner) that help regenerated axons reach their intended targets [18].

Schwann cells play a crucial role after Wallerian degeneration, beginning to proliferate and gradually enhancing the expression of GFAP and S100β, which are proteins expressed by non-myelinating and myelinating Schwann cells, respectively [19,20,21,22]. GFAP, which is the cytoskeletal constituent of Schwann cells, is expressed by immature, dedifferentiated non-myelinating cells and mature non-myelinating cells. Peripheral nerve damage causes Schwann cells to lose contact with axons, upregulating GFAP expression and developing an immature, dedifferentiated phenotype similar to that of non-myelin-forming Schwann cells [15]. Similar to previously reported data from in vitro studies, our results revealed a direct interaction between BoNT/A and Schwann cells [23]. The interaction between Schwann cells in damaged nerves is also essential for creating a beneficial environment for axon maturation, sprouting, and regrowth [24]. In our study, significantly higher expression levels of GAP43 and NF200 were observed in the BoNT/A groups than in the control group. GAP43 and NF200 expression can be used to identify the reactivation of neural regeneration programs [25]. BDNF and NGF stimulate axonal regeneration in dorsal root ganglion neurons [26]. The expression of BDNF and NGF showed similar trends to the other proteins studied. All of these results suggest that the 3.5 and 7.0 U/kg BoNT/A groups showed a superior nerve regeneration ability compared to the 14 U/kg BoNT/A and control groups. 

There are several methods for injecting BoNT/A (including intramuscular, perineural, and intraneural) [3,7,16]. In this study, we attempted to determine the effect of BoNT/A on nerve regeneration, and intramuscular injection was judged to be inappropriate for functional evaluation. Perineural injection was also judged to be inappropriate because BoNT/A could not be placed in the desired location for a long period. Intraneural injection has the advantage of being able to accurately place BoNT/A at the desired location. A previous study showed that, after the complete crushing of peripheral nerves, a single intra-nerve injection of BoNT/A increased the number of regenerated myelin fibers and the rate of axonal elongation [7]. In contrast to trials of other injectants, like local anesthetics, a previous study showed that the direct intraneural injection of BoNT/A generated minimal harm to nerves [27]. Furthermore, no rats were lost during the study due to death or significant adverse events following intraneural BoNT/A injections.

Very little research has been conducted on intraneural injections of BoNT/A. Previous studies have analyzed the effects of different concentrations of BoNT/A using different injection methods. Subcutaneous injections at concentrations of 30 U/kg have been reported to render rats lethargic [28], and intramuscular injections of high doses of BoNT/A have been reported to impair skeletal muscle function and destroy structure [29]. There is a report that the direct intraneural injection of BoNT/A is not neurotoxic, but the study only used 3.5 U/kg of BoNT/A [27]. Whether it is not toxic at higher concentrations and the mechanism of its inhibitory effect on nerve regeneration needs to be elucidated. Additional tests such as the MTT assay in Schwann cells, which measures cell viability and proliferation, may help to explain why intraneural injections of high concentrations of BoNT/A contribute less to nerve regeneration.

Previous studies were reviewed to establish appropriate comparative doses of BoNT/A. The study of Cobianchi used BoNT/A (15 pg) in 2 μL of saline intraneurally, which corresponds to a concentration of approximately 10.9 U/kg [7]. This study reported that BoNT/A accelerates sensorimotor recovery by stimulating myelinated axonal regeneration. Seo et al. used 7 U/kg of BoNT/A to report successful nerve regeneration compared to a control group [10]. Based on earlier investigations showing minimal side effects and great effectiveness at low doses, a single session of intra-nerve 7.0 U/kg BoNT/A was set as the median value in this study. According to our experimental hypothesis, the 3.5 and 14 U/kg BoNT/A groups were set up for comparison. 

This study had some limitations. First, our results were based on animal studies. It is crucial to remember that the BoNT/A doses used in animal models cannot be translated into therapeutic doses for humans based on weight ratios; rather, they must be carefully selected based on toxicity factors [8]. Furthermore, it would be of considerable interest to extend these studies to larger animals, where the magnitude of change observed in rodents may have more profound consequences owing to their longer peripheral nerves and slower rate of spontaneous regeneration. Second, there are more factors to consider when injecting BoNT/A, based on the aforementioned study on the use of BoNT/A causing injury, serotype/subtype selection, and the interval between toxin injection and injury. Our study used onabotulinum neurotoxin A, which means that the dosage we recommend cannot be extrapolated to other brands or other serotypes. In particular, it is difficult to perform a precise intraneural injection of BoNT/A without an opening. Therefore, perineural injections may be more suitable for clinical use. In addition, new preclinical studies on other injection methodologies and the timing of injections are required. As for timing, we expect that injecting BoNT/A immediately after nerve injury will be more effective than delaying it. After the point of proliferation of Schwann cells following Wallerian degeneration, we believe that injections of BoNT/A that enhance the activity of Schwann cells will not have a significant impact. However, patients do not visit the clinic immediately after injury, so further studies are definitely needed, including how delayed injection differs in outcomes from no treatment or normal saline injection. Third, because we observed favorable neural regeneration at the lowest concentration, further research on the effectiveness of lower concentrations is also warranted. Finally, there have been no clear reports of BoNT/A to injure nerves when injected intraneurally, but further research is needed to determine if there is a difference in the mechanism of action between BoNT/A injections intraneurally and other methodologies.

## 4. Conclusions

These results implied better nerve regeneration in all BoNT/A-injected groups than in the control group. Comparing within the BoNT/A-injected groups, the 3.5 and 7.0 U/kg BoNT/A group reported better nerve regeneration than the 14 U/kg BoNT/A group. Further studies are required to clarify the molecular processes underlying these advancements. However, as the 3.5 and 7.0 U/kg BoNT/A groups showed no statistical difference, our study recommends using 3.5 U/kg BoNT/A for its cost-effectiveness. A significant amount of further research is required to elucidate the mechanisms underlying neural regeneration.

## 5. Materials and Methods

### 5.1. Animals and Surgical Procedure

In this study, sixty-five SPF Sprague Dawley (SD) rats were used. The rats were housed in five groups of standard cages (22 × 22 × 13 cm) with unlimited access to food and water upon arrival. The temperature (23 ± 1 °C) was kept constant under a 12 h light/dark cycle (06:30–18:30). At the time of the operation, the mice were approximately six weeks old and weighed 187–213 g. Over the course of two days, the experiment took place between 13:30 and 17:30. The guiding principles of the “Guide for the Care and Use of Laboratory Animals” were followed in all aspects of animal experimentation (Institute for Laboratory Animal Research, Committee for the Update of the Guide for the Care and Use of Laboratory Animals, National Research Council of The National Academies, Washington, DC, USA; The National Academies Press: Washington, DC, USA, 2011) and approved by the “Institutional Animal Care and Use Committee” of Dongguk University (protocol code 202209239; approval date: 30 September 2022).

Surgery was carried out under the influence of 1–2% isoflurane. A skin incision was made beneath the hip to expose the right sciatic nerve at the mid-thigh level. Fine surgical scissors and forceps were then used to bluntly cut the muscle. After that, the sciatic nerve was crushed 1 cm above the division of its tibial, common peroneal, and sural branches. The sciatic nerve was crushed for 30 s using a Halsey needle holder (AE 064/13, NOPA, Tuttlingen, Germany) [30]. After applying VICRYL 3-0 (W9114, Ethicon LLC, Bridgewater Township, NJ, USA) to seal the wound, the rats were returned to their heated cages until all reflexes returned to normal.

The sciatic nerve-crushed rats were divided into five groups: (1) rats (*n* = 15) injected with 3.5 U/kg BoNT/A through the crushed sciatic nerve location immediately before the incision was sutured; (2) rats (*n* = 15) injected with 7.0 U/kg BoNT/A through the crushed sciatic nerve location immediately before the incision was sutured; (3) rats (*n* = 15) injected with 14.0 U/kg BoNT/A through the crushed sciatic nerve location immediately before the incision was sutured; (4) rats (*n* = 15, control) injected with 0.9% saline (70 μL/kg) after sciatic nerve injury; (5) rats (*n* = 5, sham) after exposing the sciatic nerve whose skin and muscle were sutured immediately. At weeks 3, 6, and 9, sciatic nerves were harvested from all groups, except the sham group. In the sham group, neurohistology was not performed, and only weekly neurophysiological and behavioral testing were performed for nine weeks. All intraneural injections were performed extrafascicularly by a medical professional to avoid potential mechanical nerve injury [31].

### 5.2. Experimental Groups

Before suturing the lesion, onabotulinum neurotoxin A (Allergan, Irvine, CA, USA) was intraneurally administered into the nerve-crush site using a 50 μL Hamilton Syringe (HAMILTON CO., NO706, Hamilton, OH, USA). The BoNT/A dose was based on our previous study that demonstrated efficient neural regeneration without side effects [10]. One hundred U of botulinum toxin is equivalent to 4.8 ngtox [28]. Stock solutions of BoNT/A (100 U/mL) were freshly prepared by dilution in 0.5 mL of normal saline (0.9% NaCl). To compare the effect of neural regeneration, several doses of intra-nerve BoNT/A (3.5 U/kg, 168 pgtox/kg body weight; 7.0 U/kg, 336 pgtox/kg body weight; 14 U/kg, 672 pgtox/kg body weight) were used. A single session of intraneural BoNT/A was administered as planned.

### 5.3. Neurophysiology Test

Serial electrophysiological studies were performed weekly for nine weeks. The CMAP amplitude was measured in animals under deep anesthesia (1–2% isoflurane). Single pulses of a 0.3 ms duration were applied to a pair of bar electrodes positioned at the sciatic notch to percutaneously stimulate the sciatic nerve. Microneedle electrodes were used to record the CMAP of the gastrocnemius medialis muscle. The insertion site of the Achilles tendon served as the reference electrode location. To measure the amplitude, all potentials were amplified and presented on an electromyography machine (Synergy; Viasys Healthcare, PA, USA) at the appropriate settings. Once supramaximal stimulation was reached, the CMAP amplitudes were recorded by progressively increasing the stimulus intensity [32].

### 5.4. Behavioral Tests

The gradual recovery of motor function was observed by analyzing individual gait patterns. The SFI was calculated by measuring multiple footstep parameters. The hind paws of the rats were dipped in black ink and allowed to walk along a Perspex runway corridor (12 × 12 × 60 cm) to record their footsteps. A minimum of five footsteps were recorded on each of three different tracks to calculate footstep parameters. The rats were examined weekly from weeks 1 to 9. An analysis was performed on the steps taken to measure the distances between the heel and third toe (print length, PL), the first and fifth toes (toe spread, TS) and the second and fourth toes (intermediate toe spread, IT). For both left and right footsteps, the experimental (E) and normal (N) data were recorded. The SFI was evaluated using the following equation: [SFI = −38.3 × (EPL-NPL)/NPL + 109.5 × (ETS-NTS)/NTS + 13.3 × (EIT-NIT)/NIT − 8.8] [33]. The SFI of a normal rat typically hovers around 0, whereas that of a seriously damaged rat is nearly −100.

### 5.5. Immunostaining of Sciatic Nerves

The rats were anesthetized at weeks 3, 6, and 9 using 1–2% isoflurane (Sigma-Aldrich, St. Louis, MO, USA). Distal to the crushed nerve location, a 10 mm section of the sciatic nerve was cut. The base mold was filled with a cryo-embedding medium (OCT compound), and the entire tissue was placed inside. To ensure that the tissue was completely frozen, the base mold containing the tissue block was submerged in liquid nitrogen until it reached the top of the mold. Until it was ready to section the frozen tissue block, it was kept at a temperature of −80 °C.

For immunofluorescence staining, 8 μm thick frozen tissue sections were used, and they were then transferred onto glass slides. After 30 min of room-temperature drying, the tissue sections were fixed for 1 h at 4 °C using a 4% paraformaldehyde solution (Sigma-Aldrich, Milan, Italy). The tissues upon which the antibodies were applied were permeabilized using a 0.5% Triton X-100 solution. Afterwards, the fixed sections were rinsed in phosphate-buffered saline (PBS) to achieve blocking. The slides were incubated in a 5% bovine serum albumin solution (Sigma-Aldrich, Milan, Italy) at room temperature for an hour after the fixed sections were washed in PBS. The sections were incubated with diluted primary antibodies overnight at 4 °C in a humidified room. The antibodies were purchased from Cell Signaling Technology, Inc. (Danvers, MA, USA). BoNT/A’s influence on the activity of Schwann cells in an injured sciatic nerve was evaluated through the expression of two proteins, GFAP and S100β, which are expressed by non-myelinating and myelinating Schwann cells, respectively [15]. Axonal regrowth was assessed based on the expression of GAP43, NF200, BDNF, and NGF [7,26,34,35]. The primary antibodies utilized were GFAP (1:500 dilution), S100β (1:200 dilution), GAP43 (1:250 dilution), NF200 (1:500 dilution), BDNF (1:250 dilution), and NGF (1:250 dilution). Secondary antibodies (anti-mouse IgG and anti-rabbit IgG2 at 1:500–1:1000 dilution) that had been diluted were applied to the slides the following day, rinsed with PBS, and incubated for an hour at room temperature in a humid environment. The samples were rinsed in PBS, coverslipped, and observed under a confocal microscope (STELLARIS, Leica Microsystems, Wetzlar, Germany).

### 5.6. ELISA

Using an ELISA kit (CUSABIO, Houston, TX, USA) the levels of GFAP, S100β, GAP43, NF200, BDNF, and NGF were assessed in order to evaluate the expression of proteins linked to nerve regeneration in the frozen sciatic nerve tissue that was collected. The manufacturer’s instructions were followed.

### 5.7. Toluidine Blue Staining

One 5 mm sample from each naive nerve was fixed for 48 h at 4 °C using a mixture of 2% glutaraldehyde (16216, EMS), 4% paraformaldehyde (15754-S, EMS), and 0.1 M Sorensen’s phosphate buffer pH 7.2 solution. The specimens were embedded in Araldite1502 resin (Polyscience, IL, USA) after being post-fixed in 2% osmium tetroxide and dehydrated using increasing alcohol series (starting at 50%). Using an Ultracut E microtome (Reichert Inc., Buffalo, NY, USA), semi-thin (1 μm) tissue sections were cut and stained with 1% toluidine blue. Five sample sites per section were randomly selected at 2 mm intervals on the microscope stage. These sites were examined at 1000× magnification under an optical microscope (Eclipse Ci-L, Nikon, Tokyo, Japan), and axonal regeneration was estimated based on (1) the average area of regenerated nerves, (2) the mean diameter of the myelin sheath, and (3) the axon to fiber diameter ratio (G-ratio).

For analysis, non-overlapping micrograph segments were imported into the ImageJ software (FIJI Package, version 2.0, NIH, Bethesda, MD, USA). Axons touching the borders were included in the measurements. Thresholding was performed automatically to create a black-and-white image. For fibers where the myelin touched, lines were drawn between them so that they could be segmented individually. For particle analysis, the minimum size and circularity were set at 2 μm^2^ and 0.2, respectively, to remove small motes from statistics

### 5.8. Statistical Analysis

In both graphic and textual forms, the results are presented as the mean values ± standard error of the mean or standard deviation. Using GraphPad Prism version 9 for Windows, a one-way ANOVA and Tukey’s multiple comparison test were used to assess differences between the groups at each time point (GraphPad Prism 5.01, San Diego, CA, USA). The threshold for statistical significance was a *p*-value < 0.05.

## Figures and Tables

**Figure 1 toxins-15-00691-f001:**
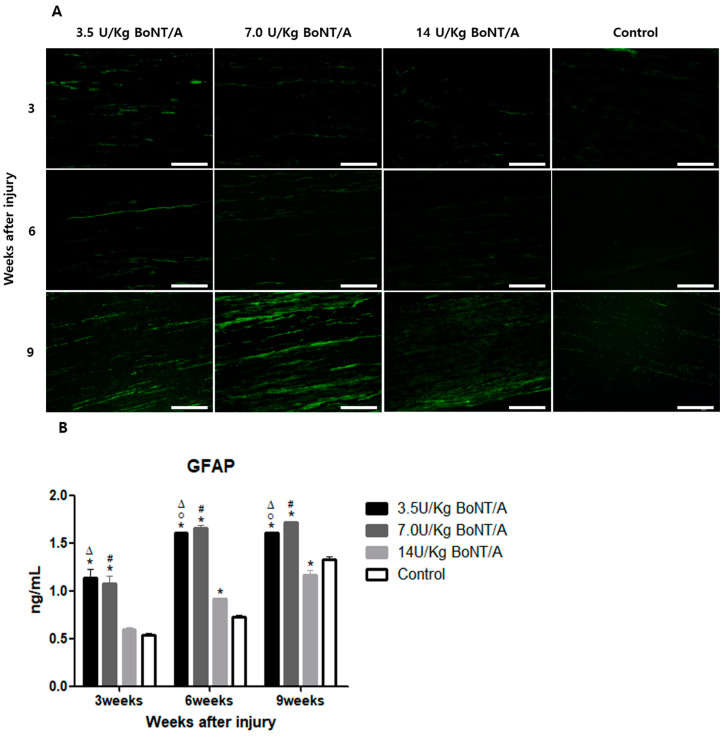
Effect of BoNT/A on the expression of glial fibrillary acid protein (GFAP) in injured sciatic nerves: (**A**) Image of immunofluorescence staining (scale bar = 150 μm, 200×). GFAP expression was higher in all BoNT/A groups (3.5, 7.0, and 14 U/kg) than in the control group. (**B**) ELISA. GFAP expression was significantly increased in the 3.5 and 7.0 U/kg BoNT/A groups compared to the 14.0 U/kg BoNT/A and control groups at all experimented weeks (*p* < 0.05). * indicates experimental group vs. control group, ° indicates 3.5 U/kg BoNT/A group vs. 7.0 U/kg BoNT/A group, Δ indicates 3.5 U/kg BoNT/A group vs. 14 U/kg BoNT/A group, and # indicates 7.0 U/kg vs. 14 U/kg BoNT/A group (*p* < 0.05). *n* = 4 in each group. The means ± standard error of the mean are shown with error bars.

**Figure 2 toxins-15-00691-f002:**
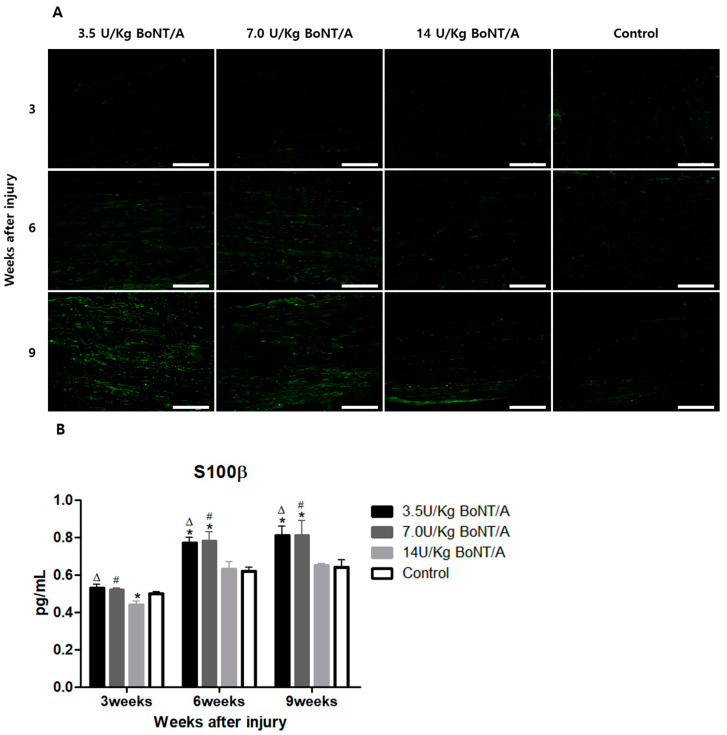
BoNT/A’s effect on the expression of astroglial calcium-binding protein S100β (S100β) in injured sciatic nerves: (**A**) Image of immunofluorescence staining (scale bar = 150 μm, 200×). Greater levels of S100β expression were noted in the 3.5 and 7.0 U/kg BoNT/A groups. (**B**) ELISA. S100β expression was significantly upregulated in the 3.5 U/kg and 7.0 U/kg BoNT/A groups at three, six, and nine weeks (*p* < 0.05). However, the 14 U/kg BoNT/A group did not show a significant difference from the control group. There was no significant difference between the 3.5 U/kg and 7.0 U/kg BoNT/A groups. * indicates experimental group vs. control group, Δ indicates 3.5 U/kg BoNT/A group vs. 14 U/kg BoNT/A group, and # indicates 7.0 U/kg BoNT/A group vs. 14 U/kg BoNT/A group (*p* < 0.05). *n* = 4 in each group. The means ± standard error of the mean are shown with error bars.

**Figure 3 toxins-15-00691-f003:**
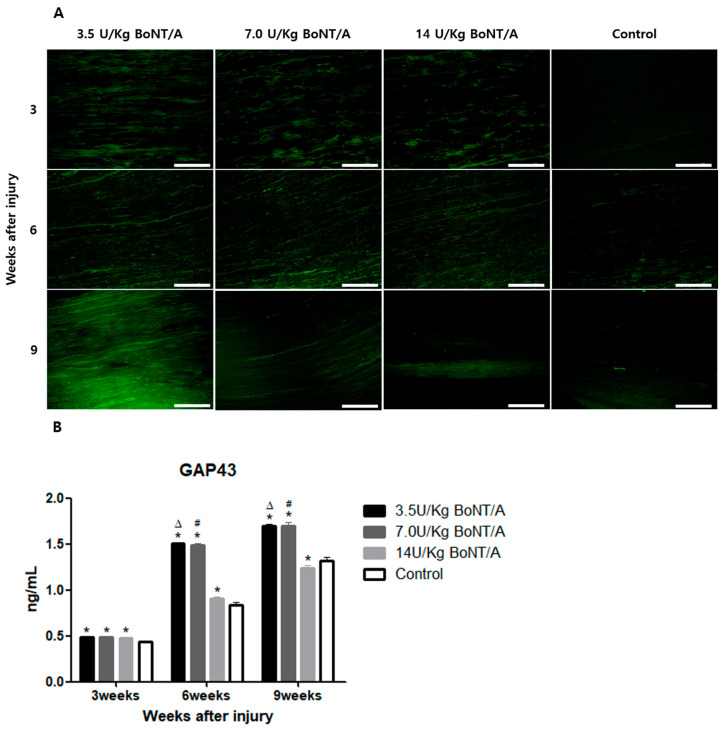
BoNT/A’s effect on growth-associated protein 43 (GAP43) expression in injured sciatic nerves: (**A**) Image of immunofluorescence staining (scale bar = 150 μm, 200×). Higher GAP43 expression was observed in all BoNT/A groups (3.5, 7.0, and 14 U/kg). The 3.5 U/kg BoNT/A group expressed especially higher levels. (**B**) ELISA. The expression levels of the 3.5 and 7.0 U/kg BoNT/A groups showed a significant increase at weeks 6 and 9 (*p* < 0.05). However, there was no significant difference between the 3.5 and 7.0 U/kg BoNT/A groups. Expression levels in the 14 U/kg BoNT/A group were significantly lower than in the control group at week 9. * indicates experimental group vs. control group, Δ indicates 3.5 U/kg BoNT/A group vs. 14 U/kg BoNT/A group, and # indicates 7.0 U/kg BoNT/A group vs. 14 U/kg BoNT/A group (*p* < 0.05). *n* = 4 in each group. The means ± standard error of the mean are shown with error bars.

**Figure 4 toxins-15-00691-f004:**
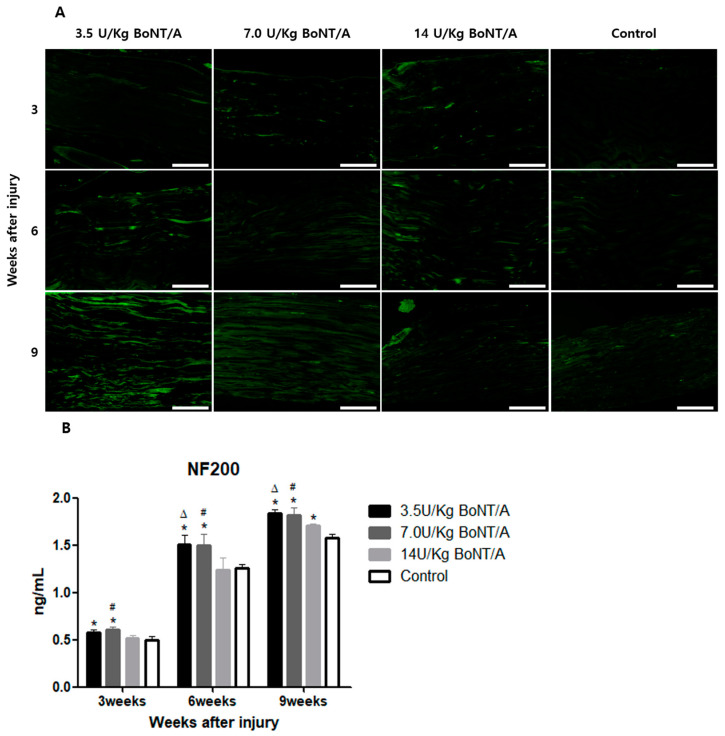
BoNT/A’s effect on neurofilament 200 (NF200) expression in injured sciatic nerves: (**A**) Image of immunofluorescence staining (scale bar = 150 μm, 200×). Higher NF200 expression levels were observed in the 3.5 U/kg and 7.0 U/kg BoNT/A groups. (**B**) ELISA. The expression levels of the 3.5 and 7.0 U/kg BoNT/A groups were significantly higher than those of the 14 U/kg BoNT/A and control groups (*p* < 0.05). No significant difference was observed between the 3.5 and 7.0 U/kg BoNT/A groups. The 14 U/kg BoNT/A group was significantly elevated compared to the control group only at week 9. * indicates experimental group vs. control group, Δ indicates 3.5 U/kg BoNT/A group vs. 14 U/kg BoNT/A group, and # indicates 7.0 U/kg BoNT/A group vs. 14 U/kg BoNT/A group (*p* < 0.05). *n* = 4 in each group. The means ± standard error of the mean are shown with error bars.

**Figure 5 toxins-15-00691-f005:**
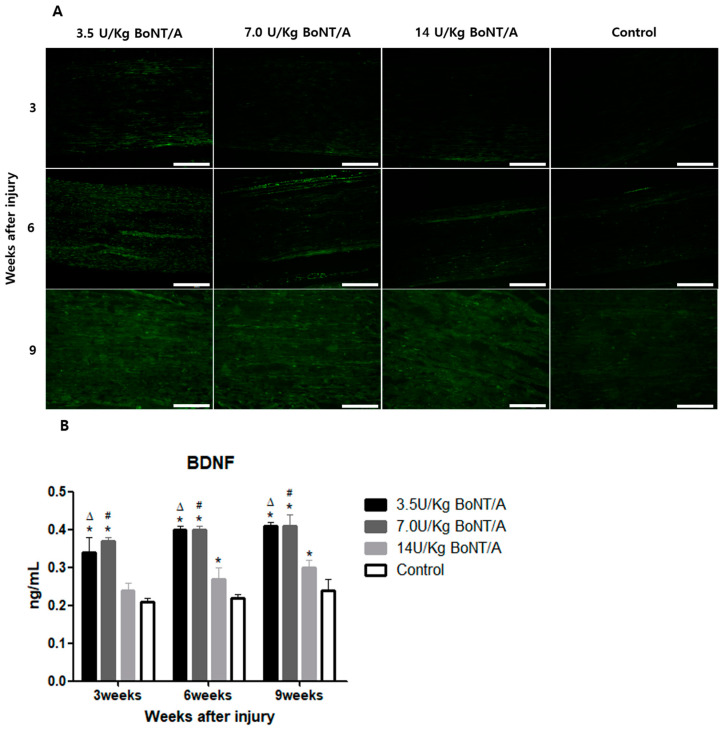
BoNT/A’s effect on the expression of brain-derived neurotrophic factor (BDNF) in injured sciatic nerves: (**A**) Image of immunofluorescence staining (scale bar = 150 μm, 200×). Higher BDNF expression levels were observed in all BoNT/A groups (3.5, 7.0, and 14 U/kg). (**B**) ELISA. BDNF expression significantly increased in the 3.5 and 7.0 U/kg BoNT/A groups compared to that in the control group at weeks 3, 6, and 9 (*p* < 0.05). At week 3, the expression in the 14 U/kg BoNT/A group was higher than that in the control group, but the difference was not statistically significant. * indicates experimental group vs. control group, Δ indicates 3.5 U/kg BoNT/A group vs. 14 U/kg BoNT/A group, and # was 7.0 U/kg BoNT/A group vs. 14 U/kg BoNT/A group (*p* < 0.05). *n* = 4 in each group. The means ± standard error of the mean are shown with error bars.

**Figure 6 toxins-15-00691-f006:**
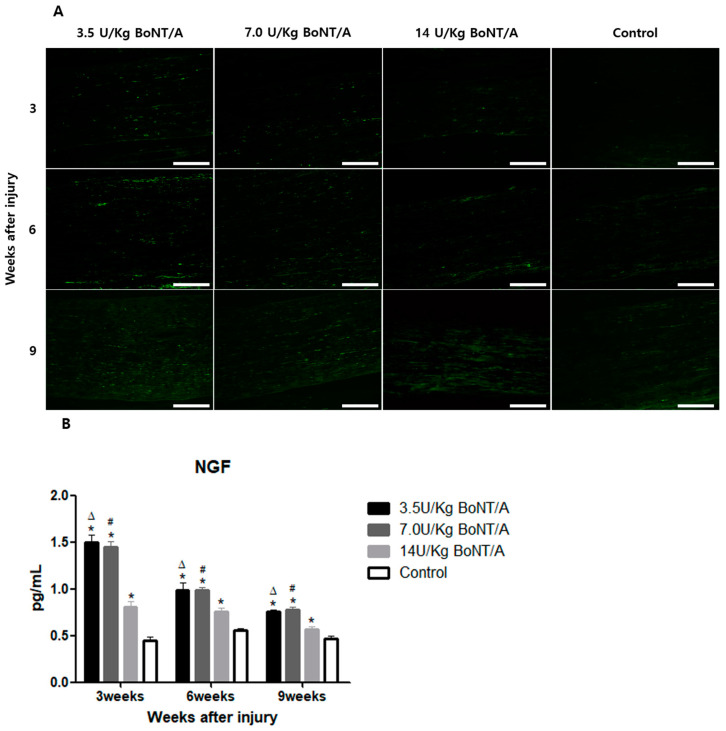
BoNT/A’s effect on nerve growth factor (NGF) expression in injured sciatic nerves: (**A**) Image of immunofluorescence staining (scale bar = 150 μm, 200×). Higher NGF expression was observed in all BoNT/A groups (3.5, 7.0, and 14 U/kg). (**B**) ELISA. NGF expression in all BoNT/A (3.5 U, 7.0 U, 14 U/kg) groups was significantly higher than that in the control group at all experimental weeks. The 3.5 and 7.0 U/kg BoNT/A group showed significantly higher expression than the 14 U/kg BoNT/A group at weeks 3, 6, and 9. * indicates experimental group vs. control group, Δ indicates 3.5 U/kg BoNT/A group vs. 14 U/kg BoNT/A group, and # indicates 7.0 U/kg vs. 14 U/kg BoNT/A group (*p* < 0.05). *n* = 4 in each group. The means ± standard error of the mean are shown with error bars.

**Figure 7 toxins-15-00691-f007:**
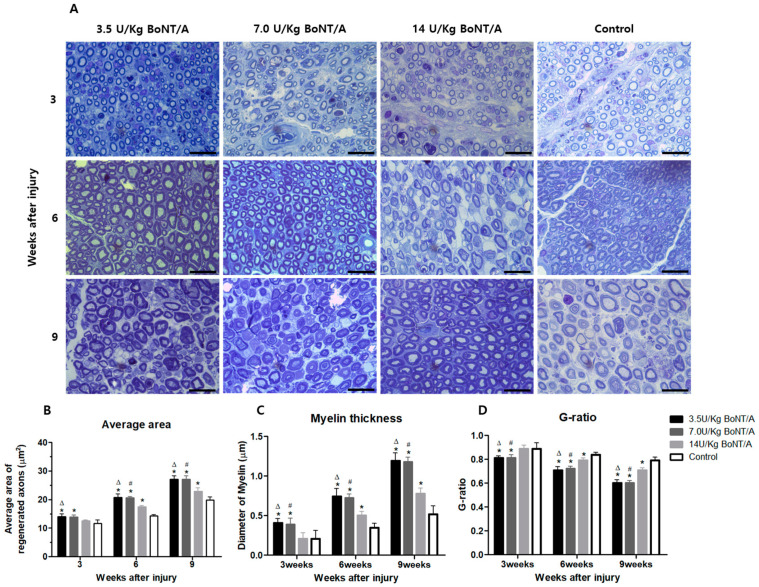
Effect of BoNT/A on axonal regeneration and remyelination: (**A**) Images of embedded sciatic nerve sections stained with toluidine blue (scale bar = 20 μm; 1000×). (**B**) Average area, (**C**) mean diameter of the myelin sheath, and (**D**) remyelination degree of the regenerated axons (G-ratio). (**B**) The average area of regenerated axons was significantly larger in the 3.5 and 7.0 U/kg BoNT/A groups than in the 14 U/kg BoNT/A and control groups. The 14 U/kg BoNT/A group had a larger area than the control group after week 6. (**C**) Comparing the mean diameter of the myelin sheath, those of the 3.5 and 7.0 U/kg BoNT/A groups were greater than that of the 14 U/kg BoNT/A and control groups. The 14 U/kg BoNT/A group was significantly different from the control group at weeks 6 and 9. (**D**) The G-ratio was significantly lower in the 3.5 and 7.0 U/kg BoNT/A groups than in the 14 U/kg BoNT/A and control groups. No significant differences were seen between the 3.5 and 7.0 U/kg groups for any of the parameters. * is experimental group vs. control group, Δ is 3.5 U/kg vs. 14 U/kg BoNT/A group, and # is 7.0 U/kg vs. 14 U/kg BoNT/A group (*p* < 0.05). *n* = 5 in each group. The means ± standard error of the mean are shown with error bars.

**Figure 8 toxins-15-00691-f008:**
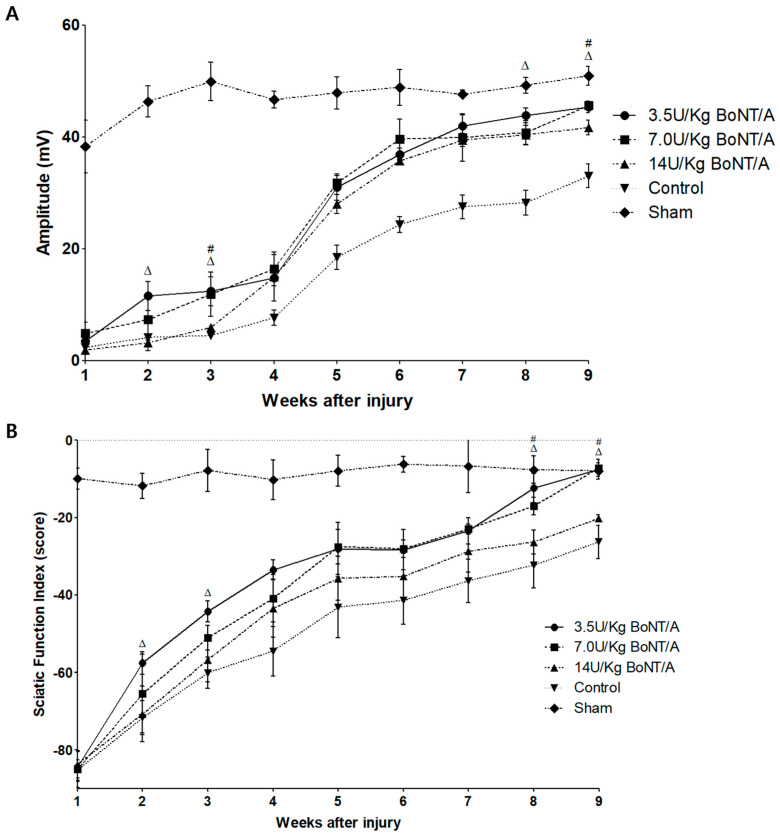
BoNT/A’s effect on functional recovery following sciatic nerve injury: For up to nine weeks after neural damage, (**A**) CMAP amplitude (mV), a representative recording of CMAP, and (**B**) SFI (score) were measured once a week (*n* = 5 per group). (**A**) Compared with the control group, all BoNT/A (3.5, 7.0, and 14 U/kg) groups showed a significant increase in amplitude after two, three, and four weeks (*p* < 0.05). Although all BoNT/A (3.5, 7.0, and 14 U/kg) groups showed significant improvement, the CMAP was still significantly lower than that in the sham group by week 9. (**B**) The 3.5 U/kg and 7.0 U/kg BoNT/A groups were statistically superior to the control group after the second and third weeks (*p* < 0.05). However, the 14 U/kg BoNT/A group was found to be significantly different from the control group only in the ninth week. SFI was not significantly different from the sham group after week 8 for the 3.5 U/kg BoNT/A group and week 9 for the 7.0 U/kg BoNT/A group. There was no significant difference between the 3.5 and 7.0 U/kg BoNT/A groups in all experimented weeks. Δ is *p* < 0.05 for 3.5 U/kg BoNT/A group vs. 14 U/kg BoNT/A group; # is *p* < 0.05 for 7.0 U/kg BoNT/A vs. 14 U/kg BoNT/A group. Statistical significance between the experimental and control groups was omitted. The means ± standard error of the mean are used to present data with error bars. *n* = 5 in each group. Δ indicates 3.5 U/kg BoNT/A group vs. 14 U/kg BoNT/A group, and # indicates 7.0 U/kg BoNT/A group vs. 14 U/kg BoNT/A group (*p* < 0.05). *n* = 5 in each group.

## Data Availability

Data are contained within the article.

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
