# Peer review of "Comparison of the Effects of Botulinum Toxin Doses on Nerve Regeneration in Rats with Experimentally Induced Sciatic Nerve Injury"

_toxins, 2023, doi:10.3390/toxins15120691_

Round 1

Reviewer 1 Report

Comments and Suggestions for Authors

An interesting, well designed, well conducted and well written article. I suggest to add a paragraph speculating with the lack of beneficial effect of the highest dose of botulinum toxin used compared with low-medium doses. 

Reviewer 2 Report

Comments and Suggestions for Authors

The authors present a novel rat study on the effects of botulinum toxin dose on sciatic nerve regeneration. They found that all doses facilitated regeneration relative to the control. The 3.5 U/kg and 7 U/kg both performed significantly better than the 14 U/kg. There were no major differences between the 3.5 U/kg and 7 U/kg groups. The study was very well designed and include functional outcome measures. The results are clearly presented, and the manuscript is well written. The authors highlight the limitations of the study and the next research steps appropriately. This study will make a meaningful contribution to the literature, and I look forward to the subsequent studies from this group.

Reviewer 3 Report

Comments and Suggestions for Authors

1.     The main weakness of this paper is a failure to discuss the discrepancy between the lack of effect of 14.0U/kg dose and efficacy of the lower doses. A key result is that the 3.5 and 7.0 U/kg doses are effective while the 14.0 dose is not, yet the paper is entirely missing any discussion of why that might be.   What is different about the 14.0 dose that it fails in this model?

2.     There are multiple marketed brands of BoNT type A available that are dosed differently.  This study used onabotulinumtoxinA only.  The manuscript should make it clear throughout (and not just in the methods section) that the dosage applies ONLY to onaBoNTA.  It cannot be extrapolated to other brands or other serotypes.

3.     The authors should avoid the use of the proprietary name “Botox” and use generic names instead.  e.g. line 264

4.     In the discussion, while there are multiple BoNT serotypes produced by clostridial bacteria, it should be noted that only A and B have been developed for clinical use.

5.     The statement in line 272 that BoNTA prevents the release of neuroactive substances from glial cells and neurons seems to contradict the main premise of the paper… that the intraneural injection of toxin following crush injury leads to release of neurotrophic factors that promote regeneration.  This needs to be clarified.

6.     Similarly, when used clinically, BoNTA poisons presynaptic neurons and causes dying back. It needs to be reconciled how BoNTA acts to damage neurons in some cases and yet promote healing in others.

7.     There is no mention of whether any adverse effects of intraneural injection were seen in their animals, only that others have shown it to be safe.

8.     The discussion states that nerve regeneration is linked to enhanced Schwann cell proliferation, transformation to repair phenotype and mast cell and macrophage activation.  Do the authors have any data on mast cell/macrophage activation in their animals?

9.     The issue of timing of injection in relation to nerve injury is mentioned only in passing, but is critical. In this study, the injection was given immediately after nerve crush, before closing the skin. Clinically, patients will not present for nerve injection at the time of nerve injury. The authors should address how timing of injection might influence response.

Comments on the Quality of English Language

The English language needs minor editing to improve the grammar/quality.

Round 2

Reviewer 3 Report

Comments and Suggestions for Authors

The changes made by the authors largely address the previous concerns.  However, i do not understand the statement added in response to comment #6:  “The reversible denervation of BoNT/A can cause muscle weakness and atrophy, which can lead to inaccurate gait analysis results to show nerve regeneration.”    The sentence does not make sense to me and needs to be revised.  

The response to #7 adds a statement that  "no rats dropped out of the study...."   The English language usage is incorrect.. researchers can remove rats from further procedures or from the analyses and rats can die, but the rats don't voluntarily "drop out" of a study.

The response to #8 should not refer to the reviewer in the statement added to the manuscript.  (" ....as the reviewer points out...."

Comments on the Quality of English Language

The manuscript needs editing to improve the English language usage.  The statement added in response to comment #6 especially does not make sense, likely because of the language barrier.
